# Associations among Alcohol Drinking, Smoking, and Nonrestorative Sleep: A Population-Based Study in Japan

**Yuichiro Otsuka [1,\*]** , **Ohki Takeshima [2]** , **Osamu Itani [1]** , **Yuuki Matsumoto [1]** and **Yoshitaka Kaneita [1]**

1 Division of Public Health, Department of Social Medicine, Nihon University School of Medicine, 30-1 Oyaguchi-Kamimachi, Itabashi-Ku, Tokyo 173-8610, Japan
2 Nihon University School of Medicine, 30-1 Oyaguchi-Kamimachi, Itabashi-Ku, Tokyo 173-8610, Japan
\* Correspondence: otsuka.yuichiro@nihon-u.ac.jp; Tel.: +81-3-3972-8111; Fax: +81-3-3972-5878

**Abstract:** Nonrestorative sleep (NRS) is a common sleep disorder. It is associated with several unhealthy lifestyle factors, such as skipping breakfast and lack of exercise. However, the associations between alcohol drinking, smoking, and NRS are unclear. This study examined the prevalence of NRS within the Japanese general population and the relationships among alcohol drinking, smoking, and NRS. We analyzed an anonymized dataset from a 2013 nationwide population survey (35,717 men and 39,911 women). NRS was assessed through a single-item question, and socio-demographic and lifestyle factors were assessed through self-reports. Multivariable logistic regression analyses were used to examine the associations between alcohol drinking, smoking, and NRS. The total prevalence of NRS was 22.2% (95% CI 21.8–22.7) in men and 23.4% (95% CI 23.0–23.8) in women. Further, we found that sleep duration and prevalence of NRS shared an inverse J-shaped relationship. Heavy alcohol drinking was significantly associated with NRS in both sexes. Short sleep duration and certain socioeconomic factors modified the effect of smoking on NRS in men. These results could be useful in the development of more effective sleep health policies to establish better sleep hygiene.

**Keywords:** healthcare; cross-sectional study; nonrestorative sleep; alcohol drinking; smoking

## 1. Introduction

Nonrestorative sleep (NRS) represents a subjective experience of feeling unrefreshed after waking up [1]. This is frequently observed in the general population and ranges from 1.4% to 42.1% across several regions and countries [2–6]. This wide range is indicative of both cross-cultural and geographical differences. In addition, few reliable and well-validated patient-reported outcome tools are available for its evaluation [7]. In Japan, the prevalence of NRS in the middle-aged population is 19.2% and 26.3% in men and women, respectively [5]. NRS has been observed in many sleep disorders such as insomnia, sleep-disordered breathing, sleep latency, narcolepsy, and short sleep duration [2,3]. It may cause functional impairments, such as poor daytime performance, sleepiness, and fatigue [8,9]. Moreover, it has been associated with several physical and mental disorders such as diabetes [10], depressive symptoms [11], and mortality [12]. Thus, resolving the concern due to NRS, in addition to other sleep disorders, is recognized as an important therapeutic goal.

Alcohol consumption and smoking are major public health problems. A meta-analysis showed that alcohol consumption is a risk factor for sleep disorders [13]. However, the same meta-analysis demonstrated that heavy drinking was not significantly associated with incident sleep disorders [13]. Regarding alcohol and NRS, a Swedish population study revealed that the frequency of alcohol consumption was positively associated with NRS [14]. In a cross-sectional study of 243,767 middle-aged Japanese individuals, moderate alcohol consumption (≥20 g/day) was negatively associated with NRS in men, but not in women [5]. Moreover, a cross-sectional study of 13,563 participants from the U.S.

aged 47–69 years reported a trend towards lower prevalence of NRS with higher alcohol consumption, whereas alcohol consumption was not related to NRS in the adjusted logistic regression model [15]. Thus, the associations between alcohol drinking and NRS are inconclusive.

Similar to alcohol, smoking is also a risk factor for sleep disorders [16]. A meta-analysis reported that smokers were 47% likelier to experience sleep problems than non-smokers [16]. However, owing to the possibility of publication bias, these estimates may be unreliable. Regarding the association between smoking and NRS, a large-scale study on middle-aged Japanese adults reported that smoking was associated with a higher chance of NRS in women, but the opposite relationship was observed in men [5]. However, other studies investigating the associations between smoking and NRS were inconclusive [15,17,18]. For example, smoking was associated with difficulty falling asleep, but not NRS [15].

Although there is growing evidence regarding factors related to NRS, previous studies in Japan have primarily targeted healthy middle-aged individuals [5,18]; thus, the prevalence of NRS among the Japanese general population is unknown. Moreover, while previous studies have suggested that unhealthy lifestyle factors are associated with NRS [3,5], few studies have examined the relationship between drinking or smoking habits and NRS scores adjusted for individual stress.

Therefore, this study aimed to examine the prevalence of NRS and identify the association among alcohol consumption, smoking, and NRS in the Japanese adult population. Considering these associations, we hypothesized that sex is a key confounding factor and decided to perform sex-stratified analyses. In this regard, it must be mentioned that significant differences in terms of sex have been observed in the prevalence of smoking and alcohol-related mortality in Japan [19,20]. Our results have key implications for the guidance and development of health policies.

## 2. Results

### 2.1. Participant Characteristics

Initially, 79,653 adults (37,571 men) were included in this study. We excluded 4025 participants for missing data on NRS. Finally, 75,628 participants (35,717 men) were included, and their data were analyzed. Table 1 shows the characteristics of the participants according to their sex. In total, 8643 participants' data on education status and 3125 participants' data on mental distress were missing due to the lack of responses. Compared with the male group, the female group had healthier eating habits, exercised less, experienced more distress, consumed less alcohol, and smoked less; they were less educated and less likely to be married.

**Table 1.** Demographic Characteristics of Analyzed Participants.

|  | Men (N = 35,717) | | Women (N = 39,911) | |
|---|---|---|---|---|
|  | N | % | N | % |
| Age class |  |  |  |  |
| 20–29 | 4013 | 11.2 | 4122 | 10.3 |
| 30–39 | 5729 | 16.0 | 6005 | 15.1 |
| 40–49 | 6362 | 17.8 | 6703 | 16.8 |
| 50–59 | 5787 | 16.2 | 6140 | 15.4 |
| 60–69 | 6971 | 19.5 | 7565 | 19.0 |
| 70–79 | 4880 | 13.7 | 5901 | 14.8 |
| 80+ | 1975 | 5.5 | 3475 | 8.7 |

**Table 1.** *Cont.*

| | | Men (N = 35,717) | | Women (N = 39,911) | |
|---|---|---|---|---|---|
| | | N | % | N | % |
| Alcohol | | | | | |
| | None | 14,157 | 39.6 | 28,191 | 70.6 |
| | ≥0, <23 g | 10,507 | 29.4 | 8539 | 21.4 |
| | ≥23, <46 g | 6379 | 17.9 | 1875 | 4.7 |
| | ≥46, <69 g (men) ≥46 g (women) | 2713 | 7.6 | 915 | 2.3 |
| | ≥69 g (men) | 1591 | 4.5 | - | - |
| | Missing | 370 | 1.0 | 391 | 1.0 |
| Smoking | | | | | |
| | Non-smoker | 23,368 | 65.4 | 35,386 | 84.1 |
| | Light smoker | 9546 | 26.7 | 3771 | 9.0 |
| | Heavy smoker | 2460 | 6.9 | 405 | 1.0 |
| | Missing | 343 | 1.0 | 349 | 0.8 |
| Mental distress | | | | | |
| | Light | 25,097 | 70.3 | 26,058 | 65.3 |
| | Moderate | 8027 | 22.5 | 10,262 | 25.7 |
| | Serious | 1281 | 3.6 | 1788 | 4.5 |
| | Missing | 1312 | 3.7 | 1803 | 4.5 |
| Health behavior | | | | | |
| | Eating regular meals | 17,996 | 50.4 | 23,345 | 58.5 |
| | Moderately exercising | 13,527 | 37.9 | 13,492 | 33.8 |
| Marital status | | | | | |
| | Unmarried | 10,898 | 30.5 | 15,085 | 37.8 |
| | Married | 24,819 | 69.5 | 24,826 | 62.2 |
| Education class | | | | | |
| | Low | 4613 | 12.9 | 5994 | 15.0 |
| | Middle | 15,810 | 44.3 | 19,403 | 48.6 |
| | High | 11,263 | 31.5 | 9902 | 24.8 |
| Missing | | 4031 | 11.3 | 4612 | 11.6 |

Participants for whom the data were missing were excluded from the analyses.

### 2.2. The Prevalence of NRS

Table 2 shows the prevalence of NRS according to the demographic characteristics. The total prevalence of NRS was 22.2% (95% CI 21.8–22.7) and 23.4% (95% CI 23.0–23.8) in men and women, respectively. For age group, the prevalence was highest in women aged 40–49 years (33.5%, 95% CI 32.3–34.6) and the lowest in men aged over 80 years (11.0%, 95% CI 9.7–12.5). Regarding alcohol consumption, a U-shaped association was observed in men, while a dose–response relationship was observed in women. That is, the prevalence of NRS increased with higher alcohol consumption in women. Regarding smoking, a dose–response relationship was observed in both male and female groups. That is, the prevalence of NRS increased with higher cigarette consumption in both men and women. Regarding mental distress, linear relationships were observed between the sexes. For serious mental distress, the prevalence was 61.6% (95% CI 58.9–64.3) and 58.4% (95% CI 56.1–60.7) in men and women, respectively. Figure 1 shows the association between sleep duration and the prevalence of NRS by sex. An inverse J-shaped association was observed between sleep duration and NRS scores. With increased sleep duration, the prevalence of NRS was lower (except for the more than 9h group). In particular, with less than 5h sleep duration, the prevalence of NRS was over 70% in both sexes. In addition, these associations were observed in the over 6h sleep duration group (Supplemental Table S1).

**Table 2.** Prevalence of Nonrestorative Sleep (NRS) by Demographic Characteristics.

| | Men (N = 35,717) | | | | Women (N = 39,911) | | | |
|---|---|---|---|---|---|---|---|---|
| | % | 95%CI | | *p*-Value | % | 95%CI | | *p*-Value |
| Total | 22.2 | 21.8 - | 22.7 | | 23.4 | 23.0 - | 23.8 | |
| Age class | | | | | | | | |
| 20–29 | 24.1 | 22.8 - | 25.4 | <0.001 | 23.7 | 22.4 - | 25.1 | <0.001 |
| 30–39 | 29.3 | 28.2 - | 30.5 | | 27.9 | 26.8 - | 29.1 | |
| 40–49 | 31.2 | 30.0 - | 32.3 | | 33.5 | 32.3 - | 34.6 | |
| 50–59 | 25.6 | 24.5 - | 26.8 | | 29.8 | 28.7 - | 31.0 | |
| 60–69 | 14.7 | 13.9 - | 15.6 | | 17.3 | 16.5 - | 18.2 | |
| 70–79 | 11.8 | 11.0 - | 12.8 | | 14.3 | 13.4 - | 15.2 | |
| 80+ | 11.0 | 9.7 - | 12.5 | | 13.3 | 12.2 - | 14.5 | |
| Alcohol/day | | | | | | | | |
| None | 22.5 | 21.8 - | 23.2 | <0.001 | 22.6 | 22.1 - | 23.1 | <0.001 |
| ≥0, <23 g | 22.5 | 21.7 - | 23.3 | | 24.6 | 23.7 - | 25.5 | |
| ≥23, <46 g | 19.9 | 19.0 - | 20.9 | | 26.3 | 24.4 - | 28.4 | |
| ≥46, <69 g (men) ≥ 46 g (women) | 22.0 | 20.5 - | 23.7 | | 32.8 | 29.8 - | 35.9 | |
| ≥69 g (men) | 28.6 | 26.4 - | 30.9 | | | | | |
| Smoking | | | | | | | | |
| Non-smoker | 21.1 | 20.5 - | 21.6 | <0.001 | 22.5 | 22.1 - | 23.0 | <0.001 |
| Light smoker | 23.7 | 22.8 - | 24.6 | | 31.0 | 29.5 - | 32.5 | |
| Heavy smoker | 27.9 | 26.2 - | 29.7 | | 35.8 | 31.1 - | 40.7 | |
| Mental distress | | | | | | | | |
| Light | 15.5 | 15.1 - | 16.0 | <0.001 | 15.8 | 15.4 - | 16.3 | <0.001 |
| Moderate | 36.6 | 35.6 - | 37.7 | | 36.9 | 35.9 - | 37.8 | |
| Serious | 61.6 | 58.9 - | 64.3 | | 58.4 | 56.1 - | 60.7 | |

Participants for whom the data were missing were excluded from the analyses. Abbreviations: CI, confidence interval; *p*-value was calculated using the χ2 test.

### 2.3. Associations among Drinking Alcohol, Smoking, and NRS

Table 3 shows the ORs of NRS scores for alcohol consumption and smoking status among men. The adjusted values significantly increased for alcohol consumption > 69 g/day (OR: 1.43, 95% CI: 1.20–1.69) and for heavy smokers (OR: 1.59, 95% CI: 1.36–1.87) when those of non-alcohol drinkers and non-smokers were used as the reference in Model 1, respectively. Models 2 and 3 showed that the adjusted ORs for NRS were significantly increased for alcohol consumption > 69 g/day (OR: 1.38, 95% CI: 1.12–1.70 in Model 2, OR: 1.31, 95% CI: 1.05–1.63 in Model 3) and 46–69 g/day (OR: 1.21, 95% CI: 1.03–1.42 in Model 2); however, smoking status was not associated with NRS in both models.

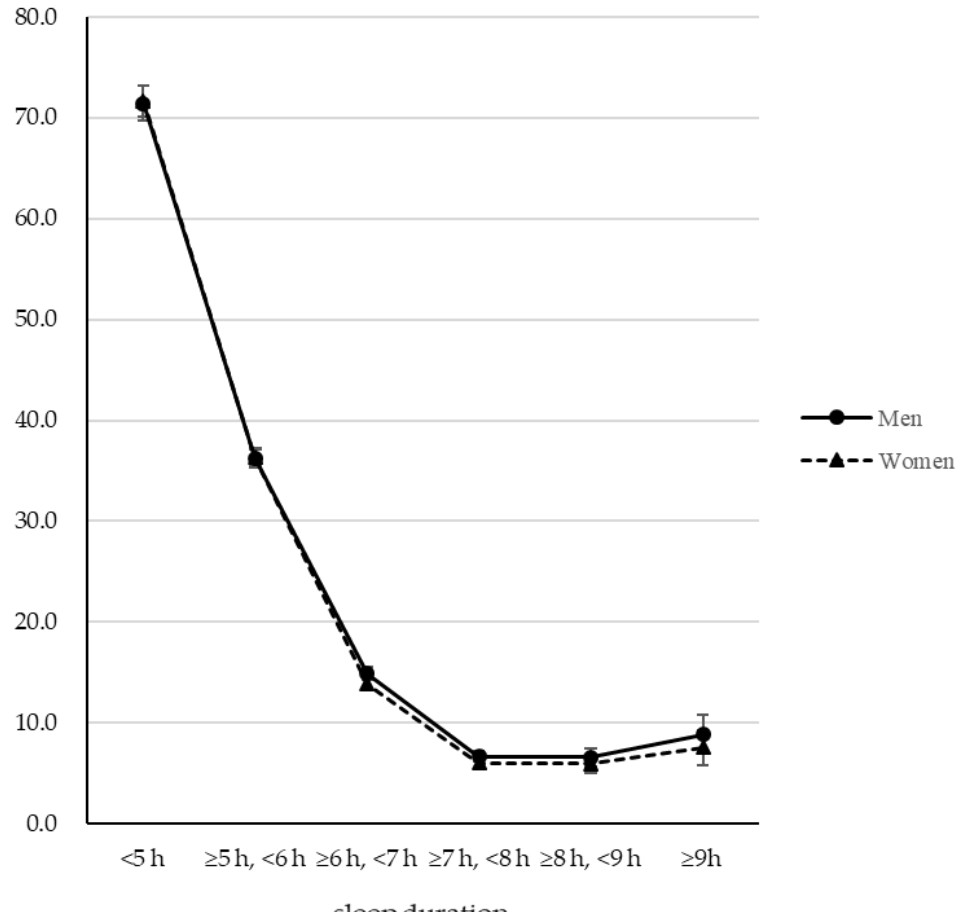

**Figure 1.** The association between sleep duration and the prevalence of NRS by sex.

**Table 3.** Odds ratio of NRS for alcohol and smoking among men.

| | Model 1 (N = 35,129) | | | | Model 2 (N = 33,903) | | | | Model 3 (N = 30,424) | | | |
|---|---|---|---|---|---|---|---|---|---|---|---|---|
| | OR | 95% CI | | *p*-Value | OR | 95% CI | | *p*-Value | OR | 95% CI | | *p*-Value |
| Alcohol/day (ref non-alcohol drink) | | | | | | | | | | | | |
| ≥0, <23 g | 1.02 | 0.95 | - | 1.10 0.565 | 1.08 | 0.99 | - | 1.17 0.104 | 1.02 | 0.93 | - | 1.12 0.735 |
| ≥23, <46 g | 0.91 | 0.83 | - | 1.00 0.055 | 1.04 | 0.93 | - | 1.16 0.505 | 0.96 | 0.86 | - | 1.08 0.525 |
| ≥46, <69 g | 1.12 | 0.98 | - | 1.29 0.097 | 1.21 | 1.03 | - | 1.42 0.019 | 1.08 | 0.91 | - | 1.28 0.384 |
| ≥69 g | 1.43 | 1.20 | - | 1.69 <0.001 | 1.38 | 1.12 | - | 1.70 0.002 | 1.31 | 1.05 | - | 1.63 0.015 |
| Smoking (ref non-smoker) | | | | | | | | | | | | |
| Light smoker | 1.05 | 0.95 | - | 1.15 0.350 | 0.96 | 0.86 | - | 1.08 0.500 | 0.97 | 0.87 | - | 1.10 0.668 |
| Heavy smoker | 1.59 | 1.36 | - | 1.87 <0.001 | 1.13 | 0.94 | - | 1.37 0.199 | 1.14 | 0.94 | - | 1.40 0.191 |

Abbreviations: NRS: nonrestorative sleep, CI: confidence interval. Participants for whom the data were missing were excluded from the analyses. Model 1: Adjusted age group and interaction between alcohol and smoking. Model 2: Model 1 + diet, exercise, sleep duration, and mental distress. Model 3: Model 2 + marital status and education class.

In the sensitivity analysis, the ORs of NRS were in the same direction as the original results (Supplemental Table S2). However, they were significantly increased for light smokers (OR: 1.24, 95% CI: 1.05–1.47) and heavy smokers (OR: 1.52, 95% CI: 1.13–2.05) when those in non-smokers were used as the reference in Model 3.

Table 4 shows the odds ratios of NRS scores for alcohol consumption and smoking status among women. The adjusted ORs for NRS in Model 1 showed a dose–response relationship with alcohol drinking status (ORs were 1.01, 95% CI: 0.95–1.07, 1.12, 95% CI: 0.98–1.26, and 1.47, 95% CI: 1.22–1.78 for those who drank alcohol 23 g/day, 23–46 g/day, and $\geq$46 g/day, respectively) and smoking status (ORs were 1.41 and 2.14 for those who smoked <20 cigarettes/day and $\geq$20 cigarettes/day, respectively). In Models 2 and 3, the adjusted odds ratios ORs for NRS were significantly increased for alcohol consumption > 46 g/day (OR: 1.44, 95% CI: 1.14–1.82 in model 2, OR: 1.36, 95% CI: 1.06–1.73 in model 3). Smoking status was not significantly associated with the NRS score. In the sensitivity analysis, the ORs of NRS were in the same direction as the original results in all the models (Supplemental Table S3). However, in Model 3, the associations among alcohol consumption, smoking status, and NRS scores were not significant.

**Table 4.** Odds ratio of NRS for alcohol and smoking among women.

| | Model 1 (N = 39,333) | | | Model 2 (N = 37,652) | | | Model 3 (N = 33,695) | | |
|---|---|---|---|---|---|---|---|---|---|
| | **OR** | **95% CI** | ***p-*Value** | **OR** | **95% CI** | ***p-*Value** | **OR** | **95% CI** | ***p-*Value** |
| Alcohol/day (ref non-alcohol drink) | | | | | | | | | |
| $\geq$0, <23 g | 1.01 | 0.95 - 1.07 | 0.814 | 0.96 | 0.89 - 1.03 | 0.238 | 0.95 | 0.88 - 1.02 | 0.170 |
| $\geq$23, <46 g | 1.12 | 0.98 - 1.26 | 0.088 | 1.11 | 0.96 - 1.29 | 0.152 | 1.11 | 0.95 - 1.29 | 0.201 |
| $\geq$46 g | 1.47 | 1.22 - 1.78 | <0.001 | 1.44 | 1.14 - 1.82 | 0.002 | 1.36 | 1.06 - 1.73 | 0.014 |
| Smoking (ref non-smoker) | | | | | | | | | |
| Light smoker | 1.41 | 1.28 - 1.56 | <0.001 | 1.02 | 0.90 - 1.15 | 0.741 | 1.04 | 0.91 - 1.18 | 0.587 |
| Heavy smoker | 2.14 | 1.61 - 2.84 | <0.001 | 0.95 | 0.66 - 1.37 | 0.782 | 1.08 | 0.72 - 1.60 | 0.720 |

Abbreviations: NRS: nonrestorative sleep, CI: confidence interval. Participants for whom the data were missing were excluded from the analyses. Model 1: Adjusted age group and interaction between alcohol and smoking. Model 2: Model 1 + diet, exercise, sleep duration, and mental distress. Model 3: Model 2 + marital status and education class.

## 3. Discussion

This is the first study to examine the association among alcohol consumption, smoking, and NRS scores in the general Japanese population. The primary findings were as follows: (1) the prevalence of NRS was over 20% in both sexes; (2) a dose–response relationship was observed among alcohol drinking, smoking, mental distress, and the prevalence of NRS; an inverse J-shaped pattern was observed between sleep duration and the prevalence of NRS; (3) heavy alcohol drinking was significantly associated with NRS in both sexes; (4) shorter sleep duration and socioeconomic factors modified the effect of smoking on NRS, in men. These findings could be valuable for developing more effective sleep guidelines in the future.

The prevalence of NRS in Japan found in our study was similar to that reported in a previous study [5]. A cross-sectional study conducted on a middle-aged population in Japan reported the prevalence of NRS as 33.3% using a similar single-item question [17]. Our data suggests that the prevalence of NRS is strongly associated with age, sleep duration, and mental distress. The prevalence of NRS in other countries varies because of the differences in how NRS is defined [2–4]. Few reliable and well-validated patient-reported outcome tools are available for its evaluation [7]. Further investigation using the same NRS assessment tools is required to compare the prevalence of NRS in various countries.

In line with this study, previous studies have reported that alcohol abuse and dependence are associated with NRS [21,22]. However, some previous studies have reported no association between alcohol consumption and NRS [15,18]. The inconsistent results may be due to adjustment for confounding factors such as sex [5], mental health, and sleep duration [3], as well as in the different classifications of alcohol consumption and drinking time, factors such as bedtime [3]. Thus, future studies should conduct a more detailed examination of drinking habits.

Several causal pathways between drinking alcohol and sleep have been proposed [23,24]. For example, a randomized control study showed that ethanol is metabolized in the latter half of the night, when sleep is shallow and fragmented, and it disrupts sleep continuity more in women than in men [23]. This insufficient sleep may accompany subacute sleep deprivation with continued heavy drinking leading to NRS. Thus, as with other sleep disorders [24], there may be a bidirectional relationship between alcohol use and NRS. Furthermore, nightcaps can exacerbate sleep-disordered breathing [25] and lead to NRS.

This study found that smoking was not a significant factor affecting the NRS of all participants in the multivariable models. However, among men with >6 h of sleep duration, smoking had a dose–response relationship with NRS. Sleep laboratory data, obtained using polysomnography, showed that smokers had shorter sleep duration, longer sleep latency, lower sleep efficiency, and more sleep apnea than non-smokers [26]. Thus, among men, sleep duration may modify the association between smoking and NRS. Previous studies from Japan and Europe reported that sex modified the association between smoking and insomnia symptoms or snoring [27,28]. For example, a large-scale cross-sectional study of European adults showed that passive smoking was related to habitual snoring [29]. Sleep outcomes or lifestyle habits, such as alcohol intake or the effects of passive smoking, might have contributed to these differing results. Previous research has also suggested that nicotine in cigarette smoke and its abrupt withdrawal contribute to sleep disturbances [30]. Thus, future studies need to survey a more detailed status of smoking, such as smoking timing and passive smoking.

This study has several limitations. First, as this study was cross-sectional, the causal relationship among alcohol drinking, smoking, and subjectively felt NRS remains to be fully clarified. Thus, future studies should use a cohort design with follow-up for NRS. Second, the single item NRS in this study lacked consistency and validity compared to the four-domain Nonrestorative Sleep Scale [31] and nine-item Restorative Sleep Questionnaire [32]. However, the NRS asked about the main symptom of "unrefreshing sleep". A single-item question is appealing as it minimizes the administrative burden in such a large population study. Therefore, our findings may provide important information for future NRS studies. Further studies are needed to develop the standard, validity, and reliability of NRS questionnaires. Third, alcohol consumption and smoking status were self-reported and not corroborated using objective measures. Thus, a judgment may exist regarding reporting bias. However, the questionnaires were created based on existing evidence by the Japanese MHLW. Additionally, the large sample size of the Japanese adult population provides important information for future NRS studies. Fourth, although we adjusted for key potential confounding variables, this dataset had no information on possible confounding factors such as sleep phase and sleep-disordered breathing [33]. Future research needs to use objective sleep measures such as polysomnography or the oxygen desaturation index. Fifth, the proportions of missing values ranged from 0.8% to 11.6% for alcohol consumption, smoking, mental distress, and education. The number of participants decreased because we performed a complete case analysis through multiple logistic regressions. Finally, the last available data that could be used were collected in 2013, which means that the data are relatively old. Consequently, the prevalence of NRS may have changed in recent years. Thus, future studies need to analyze more recent data.

Despite these limitations, the major strengths of this study include the large sample size with a nationwide adult population and the inclusion of key confounding covariates. Future studies should develop objective questionnaires to accurately identify drinking and smoking behaviors and establish the benefits of healthy drinking behaviors to prevent NRS.

In conclusion, this large-scale, cross-sectional study of adults in Japan suggests that a J-shaped relationship exists between alcohol drinking and NRS. The smoking effect on NRS may be modified by sleep duration. Despite the widely acknowledged importance of sleep and recovery, many Japanese people are dissatisfied with their sleep. Thus, the results could be useful in the development of more effective sleep health policies to establish accurate sleep hygiene. From the public health perspective, the results suggest: (1) the existence of a crosstalk relationship among smoking, drinking alcohol, and sleep, which are the pillars of Japan's health policy; (2) spreading awareness regarding the importance of sufficient sleep duration, while providing key implications that may help educate the general population about the effects of smoking and drinking habits on NRS. However, little is known about the scientifically validated strategies that are effective in improving sleep for the general population. Thus, researchers and policy makers should aim to survey the effectiveness of specific sleep hygiene recommendations in the general population. Furthermore, to compare our results with those of studies conducted in other countries, there is a need for developing a reliable and valid assessment tool for nonrestorative sleep.

## 4. Material and Methods

### 4.1. Participants

We obtained data from the Comprehensive Survey of Living Conditions (CSLC) in 2013 through the Ministry of Health, Labour, and Welfare (MHLW). The CSLC is a nationwide survey that investigates living conditions such as health, medical care, welfare, pension, and expenditure. The details of the survey can be found on the MHLW website [34]. To avoid specifying the personal information of the survey participants, the MHLW removed the living area, date of birth, and age; the data were recoded into 5-year age groups, and individuals over 90 years old were defined as the oldest group. The following households were excluded: (1) family size of eight or more; (2) single male-parent household; (3) family with two or more members in need of care; (4) family with a large age difference between husband and wife; (5) family with a small age difference between parents and children; (6) family with four or more individuals in the same age group. After the exclusion of these households, individuals with anonymized data were randomly selected. Finally, we obtained anonymized data from 79,433 individuals over the age of 20 years. This study was approved by the ethical review board of the Nihon University School of Medicine (Approval Code: P21-08-0) on 22 June 2021.

### 4.2. Measures

#### 4.2.1. Nonrestorative Sleep

NRS was assessed using a single question, "In the past month, did you feel refreshed after sleeping at night?" [5,10,18]. Responses were "very refreshed", "refreshed", "unrefreshed", or "very unrefreshed". NRS was defined when the answer was either "unrefreshed" or "very unrefreshed".

#### 4.2.2. Alcohol Drinking

Alcohol drinking status was assessed using self-administered questionnaires by collecting information on the type of beverage, frequency, and amount of consumption. We calculated alcohol consumption in grams of ethanol per day based on the responses. In Japan, "1 go Japanese sake" is approximately equivalent to 180 mL of Japanese sake (rice wine) or 23 g of ethanol and is the most common tool for measuring the amount of alcohol consumption. To calculate the frequency of alcohol consumption per week, we assigned a score to each category as follows: 1.5 for 1–2 days/week, 3.5 for 3–4 days/week, 5.5 for 5–6 days/week, and 7 for 7 days/week. Alcohol drinking status was divided into the

following categories: non-drinkers (never- and ex-drinkers), ($\geq$0 g, <23 g/day), ($\geq$23 g, <46 g/day), ($\geq$46 g, <69 g/day for men), ($\geq$69 g/day for men), and ($\geq$46 g/day for women). We set non-drinkers as a reference category.

### 4.2.3. Smoking

This survey gave four response options for smoking: (1) "Every day", (2) "Sometimes", (3) "Not smoking in the past month", and (4) "Never smoked". Those who chose options (1) or (2) were defined as current smokers. Then, the current smokers were asked about the average number of cigarettes they smoked per day ($\leq$10 cigarettes, 11–20 cigarettes, 21–30 cigarettes, or <31 cigarettes/day). Those who smoked 20 or more cigarettes/day were defined as heavy smokers, and those who smoked less than 20 cigarettes/day were defined as light smokers based on a previous study [35].

### 4.3. Covariates

Mental health was measured using the K6 [36], a six-item screening scale for assessing nonspecific psychological symptoms over the past 30 days. The six symptoms were as follows: feeling nervous, hopeless, restless or fidgety, worthless, depressed, and feeling that everything was an effort. Each item was rated on a scale of 0 (none of the time) to 4 (all the time), with total scores ranging from 0 to 24; higher scores indicated a greater tendency towards mental illness. We divided the data into three groups based on previous surveys [37,38]: severe mental health problem (the score of $\geq$13), moderate mental health problem (5–12), and light mental health problem (0–4). The Japanese version of the K6 has been validated [39]. Cronbach's $\alpha$ for the K6 in this study was 0.89.

Sleep duration was measured by considering the average sleep duration in the past month, using categorized questions: <5 h, 5–6 h, 6–7 h, 7–8 h, 8–9 h, or $\geq$9 h. We defined short sleep duration as <6 h [40,41]. Diet behavior was assessed by asking, "Do you eat regular morning, lunch, and evening meals? (yes, no)". Participants were asked, "Do you exercise moderately? (yes, no)". Marital status was categorized as married and unmarried. Educational class was defined in the following three categories: low (elementary school/junior high school), middle (high school graduates/technical or professional school), and high (over two years of college), based on International Standard Classification of Education [42].

### 4.4. Statistical Analyses

First, the participants' characteristics were described according to sex. Second, the prevalence of NRS was calculated using the demographic characteristics. Third, to explore the associations among alcohol, smoking, and NRS, we performed the logistic regression to estimate the odds ratios (ORs) and their 95% CIs by sex. The outcome measure was the NRS score. The independent variables were alcohol consumption and smoking status. Three types of logistic models were estimated: Model 1 included the interaction terms between alcohol and smoking, adjusted for age class, because alcohol drinking and smoking have interactive effects on cross-cue reactivity for those with cravings, subjective feelings of stimulation and sedation, and those of self-administration [43]; Model 2: Model 1 + diet behavior, exercise, sleep duration, and mental distress; and Model 3: Model 2 + marital status and education level. Short sleep duration is an established risk factor for NRS [3]. For the sensitivity analysis, we analyzed those who responded as having over 6 h of sleep duration (n = 47,965), using the same method that was used for measuring short sleep duration. All analyses were performed using Stata version 17.0 (Stata Corp, College Station, TX, USA). No imputations were completed for missing data in this study. All tests were two-tailed, with a significance level of $p < 0.05$.

**Supplementary Materials:** The following supporting information can be downloaded at: https://www.mdpi.com/article/10.3390/clockssleep4040046/s1, Table S1: Prevalence of NRS among non-SSD individuals by demographic characteristics; Table S2: Odds ratio of NRS for alcohol and smoking among men with non-SSD; Table S3: Odds ratio of NRS for alcohol and smoking among women with non-SSD.

**Author Contributions:** Conceptualization, methodology, writing—original draft, Y.O.; data curation, investigation, O.T.; software, visualization, O.I.; writing—reviewing and editing, Y.M.; and writing—reviewing and editing, supervision, Y.K. All authors have read and agreed to the published version of the manuscript.

**Funding:** This work was supported by the Ministry of Health, Labour, and Welfare, Government of Japan (Grant number #21FA1002).

**Institutional Review Board Statement:** The study was conducted in accordance with the Declaration of Helsinki, and approved by the Ethical Review Board of Nihon University School of Medicine (protocol code No. P-21-04 and date of approval 8 June 2021).

**Informed Consent Statement:** The need for informed consent was waived due to the retrospective nature of the study design and use of anonymized data.

**Data Availability Statement:** The data underlying this article will be shared upon reasonable request by the corresponding author.

**Acknowledgments:** We wish to thank the participants of this study and the support staff who made it possible.

**Conflicts of Interest:** The authors declare no conflict of interest.

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
