# Peer review of "Associations among Alcohol Drinking, Smoking, and Nonrestorative Sleep: A Population-Based Study in Japan"

_2624-5175, doi:10.3390/clockssleep4040046_

Round 1

Reviewer 1 Report

The study topic is interesting and novel.

To improve the scientific soundness of this manuscript, the following comments should be considered:

1. Data are quite old. As the study was carried out in 2013, the Authors should address the potential limitations resulting from this fact. Moreover, the Authors can mention "further research need" to clarify what should be done in the future.

2. Data presentation methods are well prepared and informative. A short description of scales used in the study (duration of sleep, smoking/drinking habits) should be described in more detail. This will be important for international comparisons.

3. Please consider 2-3 sentences on the practical implications of this study as well as the importance of these findings for public health in Japan. Moreover, a brief statement on how the findings from Japan are comparable to other countries will be useful.

Author Response

We appreciate the time and effort that you have dedicated to providing a thorough review of our manuscript. We are grateful for the insightful comments and have modified the manuscript based on the suggestions provided. Please find below our point-by-point responses to your comments.

Comment 1. Data are quite old. As the study was carried out in 2013, the Authors should address the potential limitations resulting from this fact. Moreover, the Authors can mention "further research need" to clarify what should be done in the future.

Response: Thank you for your comment. We have added this issue as a limitation of our study. In addition, we have mentioned the scope for future studies. (Marked Page 9, lines 300–302).

Comment 2. Data presentation methods are well prepared and informative. A short description of scales used in the study (duration of sleep, smoking/drinking habits) should be described in more detail. This will be important for international comparisons.

Response: Thank you for your suggestions regarding the data presentation methods. Accordingly, we have added more details for the descriptions of the scales used in the study (duration of sleep, smoking/drinking habits). (Marked Page 3, lines 102–106, lines 114–118, and lines 130–131).

Comment 3. Please consider 2-3 sentences on the practical implications of this study as well as the importance of these findings for public health in Japan. Moreover, a brief statement on how the findings from Japan are comparable to other countries will be useful.

Response: Thank you for your valuable suggestions. We have added some sentences regarding the practical implications of our study, the importance of our findings with regard to public health in Japan, and how the findings are comparable to those of other countries. (Marked Page 9, lines 313–322).

Reviewer 2 Report

This study is novel in examining the association between alcohol consumption, smoking, and NRS scores in the general Japanese population.

As the authors point out, a limitation of this study is that the NRS, alcohol consumption, and smoking status are all based on self-reported simple data, so it is questionable how accurately they represent the actual situation. However, the statistical analysis results of the large sample size of the Japanese adult population are interesting and should provide important information for future NRS studies.

Author Response

We wish to express our appreciation to the reviewer for their insightful comments, which have helped us to improve our manuscript.

Comment 1. As the authors point out, a limitation of this study is that the NRS, alcohol consumption, and smoking status are all based on self-reported simple data, so it is questionable how accurately they represent the actual situation. However, the statistical analysis results of the large sample size of the Japanese adult population are interesting and should provide important information for future NRS studies.

Response: Thank you for your valuable comment. Accordingly, we have added these aspects to the limitations of the study. (Marked Page 8, lines 292–293).

Reviewer 3 Report

This is a fairly interesting study, and the manuscript has been well prepared. I have some minor suggestions and comments below.

Lines 26-27: Is this the full definition of NRS? How is it commonly assessed and diagnosed?

Lines 67-68: Some more information in the introduction is required to provide a justification for the rationale behind sex-stratified analyses.

Lines 80-81: Please specify what is meant by “large age difference” and “small age difference”?

Lines 90-93: Is this a validated tool? Please provide a citation.

Line 108: “…less than 20 cigarettes per day...”

Lines 143-144: This part of the sentence is not correct English grammar “….using the same method as short sleep 143 duration is an established risk factor for NRS.” Please revise.

Lines 225-227: This part of the sentence is not correct English grammar “…..with the prevalence of NRS, and an inverse J-shaped pattern was observed between sleep duration and the prevalence of NRS….” Please revise.

Lines 235-238: This information should also be mentioned in the introduction section.

Line 257: “Polysomnography”

Line 283: “Polysomnography”

Lines 294-295: “The smoking effect on NRS may be modified….”

Author Response

We appreciate your thorough review of our manuscript. The comments have been helpful, and we have modified the manuscript based on the suggestions provided. Please find below our point-by-point responses to the your comments.

Comment 1. Lines 26-27: Is this the full definition of NRS? How is it commonly assessed and diagnosed?

Response: Thank you for your query. NRS is generally defined as the subjective feeling of being unrefreshed even after sleep. However, there are no validated tools for the diagnosis of NRS. Some researchers have attempted to form an operational definition and suggested that the main symptom is “unrefreshing sleep.” For example, the Nonrestorative Sleep Scale consists of four domains: refreshment from sleep, physical/medical symptoms of NRS, daytime functioning, and affective symptoms of NRS. We hope that this explanation clarifies your query.

Comment 2. Lines 67-68: Some more information in the introduction is required to provide a justification for the rationale behind sex-stratified analyses.

Response: Thank you for this suggestion. We have provided a citation to reflect this aspect. (Marked Page 2, lines 69–71).

Comment 3. Lines 80-81: Please specify what is meant by “large age difference” and “small age difference”?

Response: Thank you for your query. We apologize for the lack of clarity. Few households have a large age difference between the husband and wife and a small age difference between the parents and children. Thus, these households are excluded from the provided data because their inclusion might lead to the identification of individuals. The Ministry of Health, Labor and Welfare does not publish the cutoff point of the age difference. Hence, we could provide further information on this context.

Comment 4. Lines 90-93: Is this a validated tool? Please provide a citation.

Response: Thank you for this suggestion. We have provided a citation. (Marked Page 2, line 94).

Comment 5. Line 108: “…less than 20 cigarettes per day...”

Response: Thank you for pointing out this error for which we are truly apologetic. We have corrected it. (Marked Page 3, line 118).

Comment 6. Lines 143-144: This part of the sentence is not correct English grammar “….using the same method as short sleep 143 duration is an established risk factor for NRS.” Please revise.

Response: We apologize for the error. We have revised the sentence for grammar. (Marked Page 4, lines 151–154).

Comment 7. Lines 225-227: This part of the sentence is not correct English grammar “…..with the prevalence of NRS, and an inverse J-shaped pattern was observed between sleep duration and the prevalence of NRS….” Please revise.

Response: Thank you for pointing out this error. We have revised the sentence accordingly. (Marked Page 7, lines 234–237).

Comment 8. Lines 235-238: This information should also be mentioned in the introduction section.

Response: Thank you for your kind suggestion. Accordingly, we have added this information in the Introduction section as well. (Marked Page 1, lines 27–30).

Comment 9. Line 257: “Polysomnography” Line 283: “Polysomnography”

Response: Thank you for pointing out this error. We have corrected these terms. (Marked Page 8, lines 268 and 296).

Comment 10. Lines 294-295: “The smoking effect on NRS may be modified….”

Response: Thank you for your kind comment. We have corrected this point. (Marked Page 9, line 309).

Round 2

Reviewer 1 Report

The manuscript can be considered for publication in present form.

Author Response

Dear Reviewer 1,

We wish to express our appreciation to the reviewer for your insightful comment on our paper. We feel the comments have helped us significantly improve the paper.

Reviewer 3 Report

Well done on addressing my comments.

Author Response

We wish to express our appreciation to the reviewer for your insightful comment on our paper. We feel the comments have helped us significantly improve the paper.